# COVID-19 knowledge, attitudes, and practices among people vulnerable to HIV in Uganda: A cross-sectional cohort analysis

Job Kasule[1], Julius L. Tonzel[2,3], Natalie Burns[2,3], Tyler Hamby [2,3], Roger Ying [2,3*], Grace Mirembe[1], Immaculate Nakabuye[1], Hannah Kibuuka[1], Margaret Yacovone[4], Betty Mwesigwa[1], Trevor A. Crowell [2,3], for the Multinational Observational Cohort of HIV and other Infections (MOCHI) Study Group

1 Makerere University Walter Reed Program, Kampala, Uganda, 2 U.S. Military HIV Research Program, CIDR, Walter Reed Army Institute of Research, Silver Spring, Maryland, United States of America, 3 Henry M. Jackson Foundation for the Advancement of Military Medicine, Bethesda, Maryland, United States of America, 4 National Institute of Allergy and Infectious Diseases, National Institutes of Health, Rockville, Maryland, United States of America

¶ Membership of the MOCHI Study Group is provided in the Acknowledgements
* rying@global-id.org

## Abstract

### Background

People with behavioral vulnerability to HIV face barriers to healthcare engagement that may impede uptake of non-pharmaceutical and other interventions to prevent COVID-19. Understanding COVID-19 knowledge, attitudes, and practices in this population can inform disease prevention efforts during future pandemics.

### Materials and methods

From October 2022 to September 2024, we enrolled participants aged 14–55 years without HIV who endorsed recent sexually transmitted infection, injection drug use, transactional sex, condomless sex, and/or anal sex with male partners. At enrollment, we collected socio-behavioral data, including assessments of COVID-19 knowledge, attitudes, and practices. Robust Poisson regression with purposeful variable selection was used to estimate prevalence ratios with 95% confidence intervals for factors associated with COVID-19 preventive practices.

### Results

Among 418 participants, 228 (56.9%) were female, the median age was 21 years (interquartile range 19−24), and 362 (84.9%) reported sex work. Knowledge about SARS-CoV-2 transmission routes was high (95.4%) but lower for the consequences of genetic variants (48.5%−69.7%) and possibility for asymptomatic infection or transmission (66.7%−80.8%). Handwashing was practiced by 90.8% of participants

**Data availability statement:** The datasets generated and/or analyzed in support of this manuscript are available from the corresponding author upon reasonable request. Distribution of data will require compliance with all applicable regulatory and ethical processes, including establishment and approval of an appropriate data-sharing agreement. To request a minimal data set, please contact the Data Coordinating and Analysis Center (DCAC) at PubRequest@hivresearch.org and indicate the MOCHI (RV583) study along with the name of the manuscript.

**Funding:** This work was supported by agreements between the Henry M. Jackson Foundation for the Advancement of Military Medicine, Inc., and the U.S. Department of Defense (W81XWH-18-2-0040; HT9425-24-3-0004). This research was funded, in part, by the U.S. National Institute of Allergy and Infectious Diseases (AAI20052001) and National Institute of Mental Health (RF1 MH133442: Crowell).

**Competing interests:** The authors have declared that no competing interests exist.

in the preceding month, whereas mask-wearing (76.5%), avoiding symptomatic people (73.7%), and any history of COVID-19 vaccination (46.9%) were less prevalent. Males were more likely to report avoiding symptomatic people (adjusted prevalence ratio 1.16 [95% confidence interval 1.03–1.31]) and COVID-19 vaccination (1.30 [1.05–1.60]). Enrollment during the BQ.1/BQ.1.1 Omicron wave was associated with less mask-wearing (0.81 [0.67–0.99]) but more vaccination (1.59 [1.29–1.95]).

## Discussion

We observed variable COVID-19 knowledge and attitudes among Ugandan adolescents and adults with little impact on COVID-19 preventive practices. Efforts to address suboptimal uptake of disease preventive practices during this and future disease outbreaks will require more than just improving knowledge.

## Introduction

Uganda reported its first case of SARS-CoV-2 infection on 21 March 2020, and by December 2023, had a cumulative total of 171,888 cases and 3,632 deaths due to COVID-19 [1]. Early in the pandemic, Uganda's government mandated non-pharmaceutical interventions (NPIs) to reduce SARS-CoV-2 transmission. Broadly, NPIs are strategies other than biomedical interventions that an individual can utilize to minimize their risk of infection, such as wearing face masks in public, social distancing, and avoiding social gatherings [2–4]. Recommendations shifted to emphasize COVID-19 vaccines once they became available in March 2021 [5]. However, NPIs remained an important part of the public health response to mitigate COVID-19 [6].

Individual knowledge, attitudes, and practices towards COVID-19 impact adherence to NPIs and therefore are critical to addressing the pandemic. In a study from areas of Uganda with high COVID-19 prevalence, increased COVID-19-related knowledge was associated with positive attitudes regarding COVID-19 NPIs and increased adherence to COVID-19 NPIs [7]. Furthermore, a Ugandan cross-sectional survey found that although 93.9% of participants were knowledgeable about COVID-19, only 51.3% had positive attitudes towards presidential directives and Ministry of Health COVID-19 guidelines, and only 48.3% were adherent to COVID-19 NPIs [8]. Similar patterns of low NPI usage despite high knowledge have also been seen in other Ugandan populations such as healthcare workers and students [9,10].

As COVID-19 pharmaceutical interventions were developed, NPI mandates were reduced [11]. However, therapeutic interventions such as antivirals (e.g., remdesivir) and monoclonal antibodies were not part of routine clinical care, which primarily relied on supportive interventions [12]. Therefore vaccination, beginning in March 2021, became the mainstay biomedical intervention [13]. Vaccination efforts initially targeted individuals who were essential workers (e.g., healthcare workers, teachers, security personnel), ≥ 50 years of age, and with co-morbidities. Vaccines were provided free of charge at designated health facilities during outreach campaigns as part of the Uganda National Program on Immunisation [14].

Despite the broad access to vaccines, vaccination levels remained low. In a cross-sectional study conducted between 2021 and 2022, after COVID-19 vaccination recommendations had been expanded, only half of participants had received one or more COVID-19 vaccinations and less than half of those participants had received a full vaccine series [5]. Most participants believed that they had not received adequate information about COVID-19 vaccines and wanted more information regarding vaccines. Another study found that although most individuals were willing to receive the vaccine, very few had in fact received it [15]. However, both of these studies were conducted as rollout of the COVID-19 vaccine was occurring in 2021, with few studies taking place after vaccinations reached a plateau in 2022 [6].

People with behavioral vulnerabilities to HIV—such as men who have sex with men (MSM), people who engage in transactional sex, and people who inject drugs—face particularly high barriers to healthcare engagement. In many settings, they face stigma, unfair treatment, criminalization, homelessness and food insecurity, which can be barriers to healthcare engagement for disease prevention [16–21]. Furthermore, prior studies have identified multiple factors that are associated with both HIV and COVID-19 such as binge drinking, sexually transmitted infections (STIs), and poverty [22,23]. Given the unique challenges faced by people who are behaviorally vulnerable to HIV, specifically tailored preventive interventions are critical. Therefore, we examined COVID-19 knowledge, attitudes, and practices among people vulnerable to HIV in Uganda to guide the public health responses to this and potential future pandemics.

## Materials and methods

### Study population

From 14 October 2022 to 30 September 2024, the Multinational Observational Cohort of HIV and other Infections (MOCHI) study enrolled participants at the Makerere University-Walter Reed Program in Kampala, Uganda. With objectives to estimate the incidence of HIV and other STIs, the study enrolled participants aged 14–55 years, who had a negative HIV test, and had evidence of behavioral vulnerability to HIV [24]. Behavioral vulnerability to HIV was defined as self-report of one or more of the following in the 24 weeks prior to screening: (1) a newly diagnosed STI, (2) transactional sex, (3) condomless vaginal or anal intercourse with at least three different partners living with HIV or of unknown HIV status, (4) injection drug use, or (5) anal intercourse with one or more different male partners. Participants were excluded from the study if they had any significant medical conditions or substance dependence that would impair study participation, were a current or past participant in an HIV vaccine study, or were pregnant at the time of screening. These exclusion criteria were included to mimic a study population that may someday be recruited into a clinical trial of an HIV prevention product.

### Data collection

Participants completed demographic, sexual behavior, and COVID-19 surveys at enrollment. Computer-assisted self-interview was the preferred format for survey administration in order to mitigate social desirability bias. If participants were unable to complete the questionnaires by computer-assisted self-interview due to factors such as computer literacy or technical outage, the questionnaires were administered by trained study staff. Age was dichotomized as ≤24 years or >24 years. Education level was dichotomized as less than secondary education (<12 years) or secondary education or higher (≥12 years). Current occupation was categorized as sex worker, entertainment/hospitality, commerce/business/skilled trade, and other. Weekly household income was dichotomized at the 20th percentile in the analysis population (≤25000 Ugandan Shillings [Ush] or >25000 Ush). The COVID-19 wave at enrollment was dichotomized as the BQ.1/BQ.1.1 Omicron wave (31 October 2022−2 January 2023) or non-wave (13 October 2022−30 October 2022 and 3 January 2023−30 September 2024) [25]. Sex work or transactional sex was defined as reporting one's occupation as "sex work" or responding "Yes" to "Are you a person who is a sex worker (sex in exchange for things such as money, drugs, food, shelter, or transportation)." MSM were defined as male participants who indicated sex with males.

COVID-19 knowledge was assessed by asking participants to indicate whether certain statements were "True" or "False" (e.g., COVID-19 can be transmitted through coughing or sneezing). Answers were dichotomized as "Correct" or "Incorrect." COVID-19 attitudes were assessed by asking participants to respond on a 5-point Likert-scale to statements about COVID-19 risk (e.g., I am at risk for severe disease due to COVID-19). Responses were dichotomized as "Agree" ("Strongly agree" and "Somewhat agree") or "Disagree" ("Strongly disagree," "Somewhat disagree," "Neutral," or "Don't know"). COVID-19 preventive practices were assessed by asking participants if they used certain measures to prevent COVID-19 in the past month (e.g., Wearing a face mask or covering). Responses were dichotomized as "Yes" or "No." The practice of COVID-19 vaccination was assessed with the question, "Have you received a vaccine to prevent COVID-19?" For all survey items, "Don't know" was categorized as "No" or "Incorrect," and "Refuse to answer" was categorized as missing.

## Statistical approach

Descriptive statistics were reported as medians and interquartile ranges (IQRs) for continuous variables and as frequencies and percentages for categorical variables, which were compared using Chi-squared tests of independence. Univariable and multivariable robust Poisson regression were performed to estimate prevalence ratios (PRs) and adjusted prevalence ratios (aPRs) with 95% confidence intervals (CIs) for the sociodemographic, knowledge, and attitude variables potentially associated with selected preventive practice outcomes. Preventive practices were selected as outcomes if they were recommended by public health agencies and had sufficient variability in uptake to allow modeling (e.g., not universal endorsement). History of COVID-19 vaccination was also included as a preventive practice outcome given its widespread public health recommendation at the time. Independent models were fit for each selected preventive practice and purposeful variable selection was used to identify the independent variables for each multivariable model [26]. Briefly, purposeful variable selection involved first evaluating the univariable effect of each independent variable on the outcome and then selecting certain variables for multivariable analysis. Variables were sequentially added or removed to evaluate their effects as covariates or confounders, ultimately leading to a final multivariable model of purposefully selected variables potentially associated with each preventive practice outcome. Missingness in the independent variables was handled using a complete-case analysis, wherein only participants with complete data for all independent variables were included in all regression models. Statistical analyses were performed using R, version 4.3.2 (R Foundation for Statistical Computing, Vienna, Austria) [27].

## Ethical considerations

All participants provided written informed consent prior to any study procedures, including mature and emancipated minors. Participants as young as 14 years were considered eligible for inclusion because of known early sexual debut in Uganda and East Africa, which has been associated with high prevalence and incidence of HIV and other STIs [28–32]. According to local guidelines, participants aged 14−17 years with drug dependency, alcohol dependency, or a history STI were considered mature; participants aged 14−17 years who were pregnant, married, had a child, or catered for their own livelihood were considered emancipated. Consent was not sought from parents or guardians of mature or emancipated minors. Minors who were not considered mature or emancipated were not enrolled. Research was conducted in accordance with the principles described in the International Conference on Harmonization Good Clinical Practice guidelines, the Nuremberg Code, the Belmont Report, the Declaration of Helsinki, and U.S. federal regulations regarding the protection of human participants as described in 32 CFR 219 and Army Regulation 70−25. The study was approved by institutional review boards at the Makerere University School of Public Health, Kampala, Uganda; the Walter Reed Army Institute of Research, Silver Spring, Maryland, USA; and all collaborating institutions.

### Inclusivity in global research

Additional information regarding the ethical, cultural, and scientific considerations specific to inclusivity in global research is included in the Supporting Information (S1 File).

## Results

### Demographics

From 13 October 2022 to 30 September 2024, 422 people with behavioral vulnerability to HIV were enrolled, of whom 418 (99.1%) had responses for at least one preventive practice and were included in these analyses (Table 1). The median age was 21.0 years (IQR 19.0–24.0) and 238 (56.9%) were female. Most participants identified sex work as their primary occupation (289, 70.7%), and even more endorsed engaging in either sex work or transactional sex (362, 84.9%). Most males reported sex with men (131/180, 72.8%).

### COVID-19 knowledge and attitudes

Knowledge regarding COVID-19 was highly variable (Fig 1). Among the 418 participants with responses to questions about preventive practices, most knew that SARS-CoV-2 can be transmitted through coughing or sneezing (95.4%). Fewer participants knew about the consequences of SARS-CoV-2 genetic variants, evidenced by correctly identifying that SARS-CoV-2 variants can increase the chance that people are infected multiple times (69.7%) or that SARS-CoV-2 variants can decrease the effectiveness of COVID-19 vaccines (48.5%). Similarly, although many participants knew that people without symptoms may still be infected with SARS-CoV-2 and may be contagious (80.8%), fewer stated that they could have SARS-CoV-2 infection even in the absence of symptoms (66.7%).

Attitudes toward COVID-19 were also variable. Most participants believed that becoming infected poses a risk to others (90.6%), were concerned about the spread of COVID-19 in their community (82.2%), and were concerned about getting infected themselves (83.7%). Fewer participants believed that they were at risk for severe disease due to COVID-19 (45.3%)

### COVID-19 preventive practices

The most endorsed COVID-19 preventive practice taken in the past month was hand-washing, which was reported by 90.8% of participants (Fig 1). Mask wearing, avoiding people with symptoms, and avoiding touching one's face were also highly endorsed (76.5%, 73,7%, and 70.6%, respectively). Among the least endorsed practices were those with little public health support such as taking traditional medicines (48.3%), using antibiotics (37.3%), and wearing gloves (28.5%). A history of COVID-19 vaccination was reported by 46.9% of participants.

### Factors associated with COVID-19 preventive practices

The nearly universal uptake of hand-washing precluded its use as an outcome for modeling. Mask-wearing, avoiding people with symptoms, and history of COVID-19 vaccination were selected as outcomes for modeling given their widespread recommendation by health policymakers and variable uptake.

Participants with correct knowledge of asymptomatic SARS-CoV-2 transmission were more likely to endorse wearing a mask than participants with incorrect knowledge (79.2% vs 65.0%, $p = 0.007$; Fig 2A). Furthermore, participants with correct knowledge of the decreased effectiveness of COVID-19 vaccines against SARS-CoV-2 variants were more likely to endorse wearing a mask than those with incorrect knowledge (80.5% vs. 72.2%, $p = 0.047$). However, wearing a mask was not associated with any other COVID-19 knowledge statements. There were also no significant associations between correct responses to COVID-19 knowledge statements and either avoidance of people with symptoms (Fig 2B) or COVID-19 vaccination (Fig 2C).

**Table 1. Enrollment sociodemographic characteristics of participants with behavioral vulnerabilities to HIV in Uganda, overall and by self-reported practice of COVID-19 preventive practices.**

| Characteristic | Overall (N=418) | Wearing a face mask (N=417) | | | Avoiding people with symptoms (N=415) | | | COVID-19 vaccination (N=409) | | |
|---|---|---|---|---|---|---|---|---|---|---|
| | | Yes (n=319) | No (n=98) | p-value | Yes (n=306) | No (n=109) | p-value | Yes (n=192) | No (n=217) | p-value |
| Sex | | | | 0.692 | | | **0.011** | | | **0.048** |
| Male | 180 (43.1%) | 136 (75.6%) | 44 (24.4%) | | 144 (80.0%) | 36 (20.0%) | | 93 (52.5%) | 84 (47.5%) | |
| Female | 238 (56.9%) | 183 (77.2%) | 54 (22.8%) | | 162 (68.9%) | 73 (31.1%) | | 99 (42.7%) | 133 (57.3%) | |
| Man who has sex with men | | | | 0.574 | | | 0.686 | | | 0.454 |
| No | 33 (20.1%) | 26 (78.8%) | 7 (21.2%) | | 27 (81.8%) | 6 (18.2%) | | 15 (46.9%) | 17 (53.1%) | |
| Yes | 131 (79.9%) | 97 (74.0%) | 34 (26.0%) | | 103 (78.6%) | 28 (21.4%) | | 70 (54.3%) | 59 (45.7%) | |
| Missing | 16 | 13 | 3 | | 14 | 2 | | 8 | 8 | |
| Age | | | | 0.223 | | | 0.244 | | | 0.271 |
| 15–24 years | 340 (81.3%) | 256 (75.3%) | 84 (24.7%) | | 254 (74.9%) | 85 (25.1%) | | 152 (45.6%) | 181 (54.4%) | |
| 25–40 years | 78 (18.7%) | 63 (81.8%) | 14 (18.2%) | | 52 (68.4%) | 24 (31.6%) | | 40 (52.6%) | 36 (47.4%) | |
| Education Level | | | | **0.047** | | | 0.311 | | | 0.199 |
| Less than secondary | 252 (61.6%) | 183 (72.9%) | 68 (27.1%) | | 179 (71.9%) | 70 (28.1%) | | 110 (44.7%) | 136 (55.3%) | |
| Secondary or higher | 157 (38.4%) | 128 (81.5%) | 29 (18.5%) | | 120 (76.4%) | 37 (23.6%) | | 79 (51.3%) | 75 (48.7%) | |
| Missing | 9 | 8 | 1 | | 7 | 2 | | 3 | 6 | |
| Occupation | | | | 0.527 | | | 0.264 | | | 0.911 |
| Sex Worker | 289 (70.7%) | 223 (77.4%) | 65 (22.6%) | | 207 (72.4%) | 79 (27.6%) | | 132 (46.8%) | 150 (53.2%) | |
| Entertainment/ Hospitality | 18 (4.4%) | 12 (66.7%) | 6 (33.3%) | | 14 (77.8%) | 4 (22.2%) | | 10 (55.6%) | 8 (44.4%) | |
| Commerce/Business/ Skilled Trade | 43 (10.5%) | 30 (69.8%) | 13 (30.2%) | | 29 (67.4%) | 14 (32.6%) | | 19 (46.3%) | 22 (53.7%) | |
| Other | 59 (14.4%) | 46 (78.0%) | 13 (22.0%) | | 49 (83.1%) | 10 (16.9%) | | 28 (47.5%) | 31 (52.5%) | |
| Missing | 9 | 8 | 1 | | 7 | 2 | | 3 | 6 | |
| Weekly Household Income | | | | 0.750 | | | 0.422 | | | **0.007** |
| ≤25000 Ugandan shillings | 76 (18.6%) | 59 (77.6%) | 17 (22.4%) | | 58 (77.3%) | 17 (22.7%) | | 24 (32.9%) | 49 (67.1%) | |
| >25000 Ugandan shillings | 333 (81.4%) | 252 (75.9%) | 80 (24.1%) | | 241 (72.8%) | 90 (27.2%) | | 165 (50.5%) | 162 (49.5%) | |
| Missing | 9 | 8 | 1 | | 7 | 2 | | 3 | 6 | |
| Knows Anyone with COVID-19 | | | | 0.775 | | | 0.554 | | | 0.492 |
| No | 327 (78.4%) | 251 (77.0%) | 75 (23.0%) | | 238 (73.0%) | 88 (27.0%) | | 154 (47.8%) | 168 (52.2%) | |
| Yes | 90 (21.6%) | 68 (75.6%) | 22 (24.4%) | | 67 (76.1%) | 21 (23.9%) | | 38 (43.7%) | 49 (56.3%) | |
| Missing | 1 | 0 | 1 | | 1 | 0 | | 0 | 0 | |
| COVID-19 Wave | | | | **0.046** | | | 0.090 | | | **<0.001** |
| BQ.1/BQ1.1 Omicron Wave | 63 (15.1%) | 42 (66.7%) | 21 (33.3%) | | 41 (65.1%) | 22 (34.9%) | | 43 (68.3%) | 20 (31.7%) | |
| Non-Wave | 355 (84.9%) | 277 (78.2%) | 77 (21.8%) | | 265 (75.3%) | 87 (24.7%) | | 149 (43.1%) | 197 (56.9%) | |
| Sex Work or Transactional Sex | | | | 0.184 | | | 0.486 | | | 0.563 |
| No | 41 (10.2%) | 28 (68.3%) | 13 (31.7%) | | 32 (78.0%) | 9 (22.0%) | | 21 (51.2%) | 20 (48.8%) | |
| Yes | 362 (89.8%) | 280 (77.6%) | 81 (22.4%) | | 262 (73.0%) | 97 (27.0%) | | 164 (46.5%) | 189 (53.5%) | |
| Missing | 15 | 11 | 4 | | 12 | 3 | | 7 | 8 | |

Use of COVID-19 preventive practices was ascertained by asking participants, "In the past month, what measures have you taken to prevent infection from COVID-19." Vaccination was ascertained by asking participants, "Have you received a vaccine to prevent COVID-19?" The number of participants responding to each COVID-19 preventive practice varied due to non-response. Comparisons of COVID-19 preventive practice by characteristic were performed with Chi-squared test of independence. **Bolded** p-values indicate p<0.05.

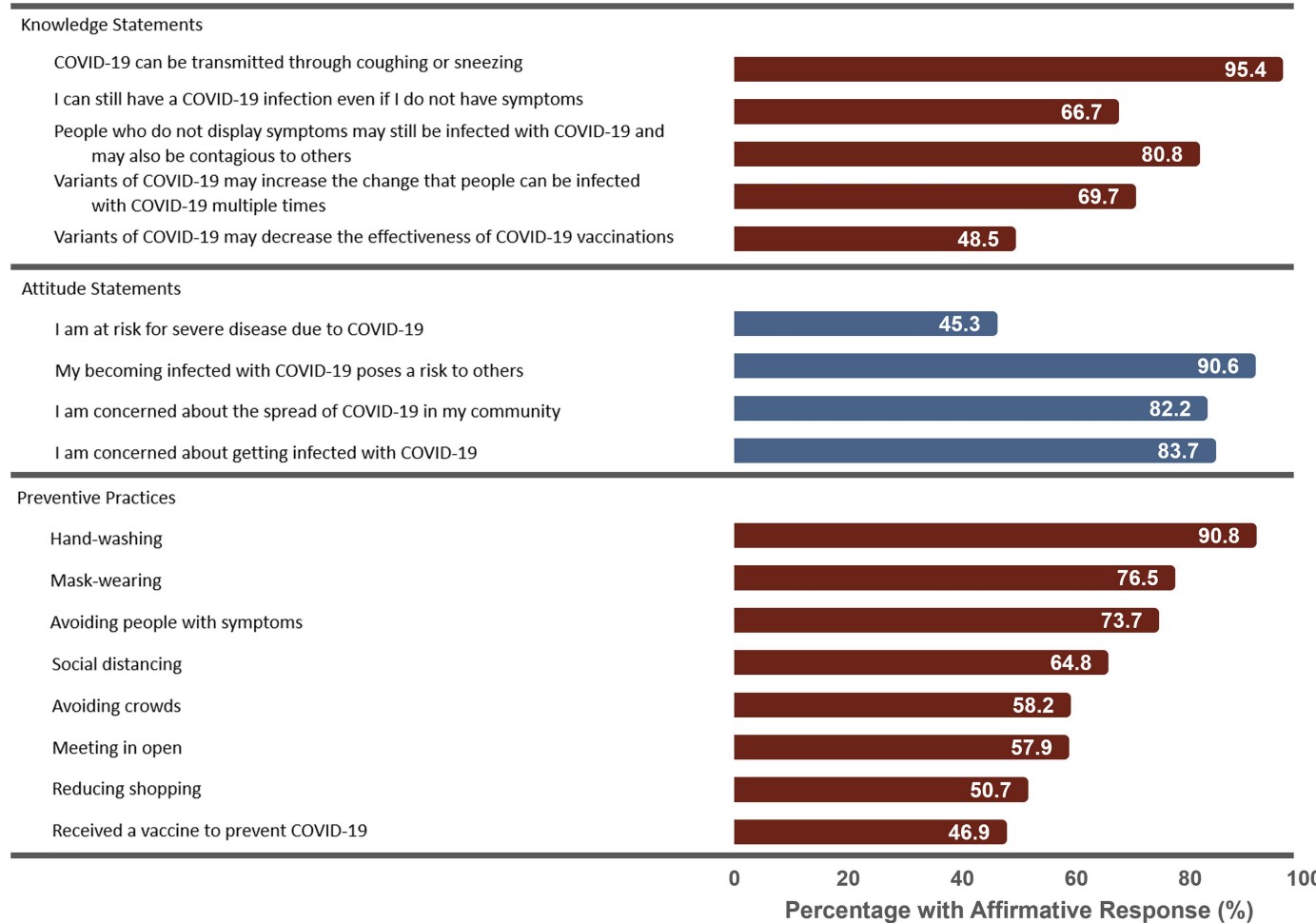

**Fig 1. Percentage of participants correctly identifying true COVID-19 knowledge statements, agreeing with COVID-19 attitude statements, and reporting COVID-19 preventive practices.** COVID-19 knowledge was ascertained by asking participants to identify statements regarding COVID-19 as True or False. COVID-19 attitudes were ascertained by asking participants their level of agreement with statements regarding COVID-19 risk on a 5-point Likert scale, with "strongly agree" or "somewhat agree" considered affirmative responses. COVID-19 preventive practices were ascertained by asking participants if they had used various non-pharmaceutical interventions to prevent COVID-19 in the past month and, for vaccination, asking, "Have you received a vaccine to prevent COVID-19?".

There was also no association between COVID-19 attitudes and wearing a mask (Fig 3A) or avoiding people with symptoms (Fig 3B). However, participants who disagreed that infection with COVID-19 poses a risk to others were more likely to report having received a COVID-19 vaccine (63.9% vs. 45.2%, p = 0.032; Fig 3C). COVID-19 vaccination was not associated with any other COVID-19 attitudes.

In multivariable analyses, most participants had complete data to be included in multivariable modeling (380/417 [91.1%] for wearing a face mask, 379/415 [91.3%] for avoiding people with symptoms, and 373/409 [91.2%] for receiving a COVID-19 vaccine). Participants who were enrolled during the BQ.1/BQ1.1 Omicron wave were less likely to wear a mask (aPR 0.81 [95% CI 0.67–0.99], p = 0.034; Table 2) and more likely to have received the COVID-19 vaccine (aPR 1.59 [95% CI 1.29–1.95], p < 0.001; Table 4). Males were more likely to avoid people with symptoms (aPR 1.16 [95% CI 1.03–1.31], p = 0.012; Table 3) and were more likely to be vaccinated against COVID-19 (aPR 1.30 [95% CI 1.05–1.60],

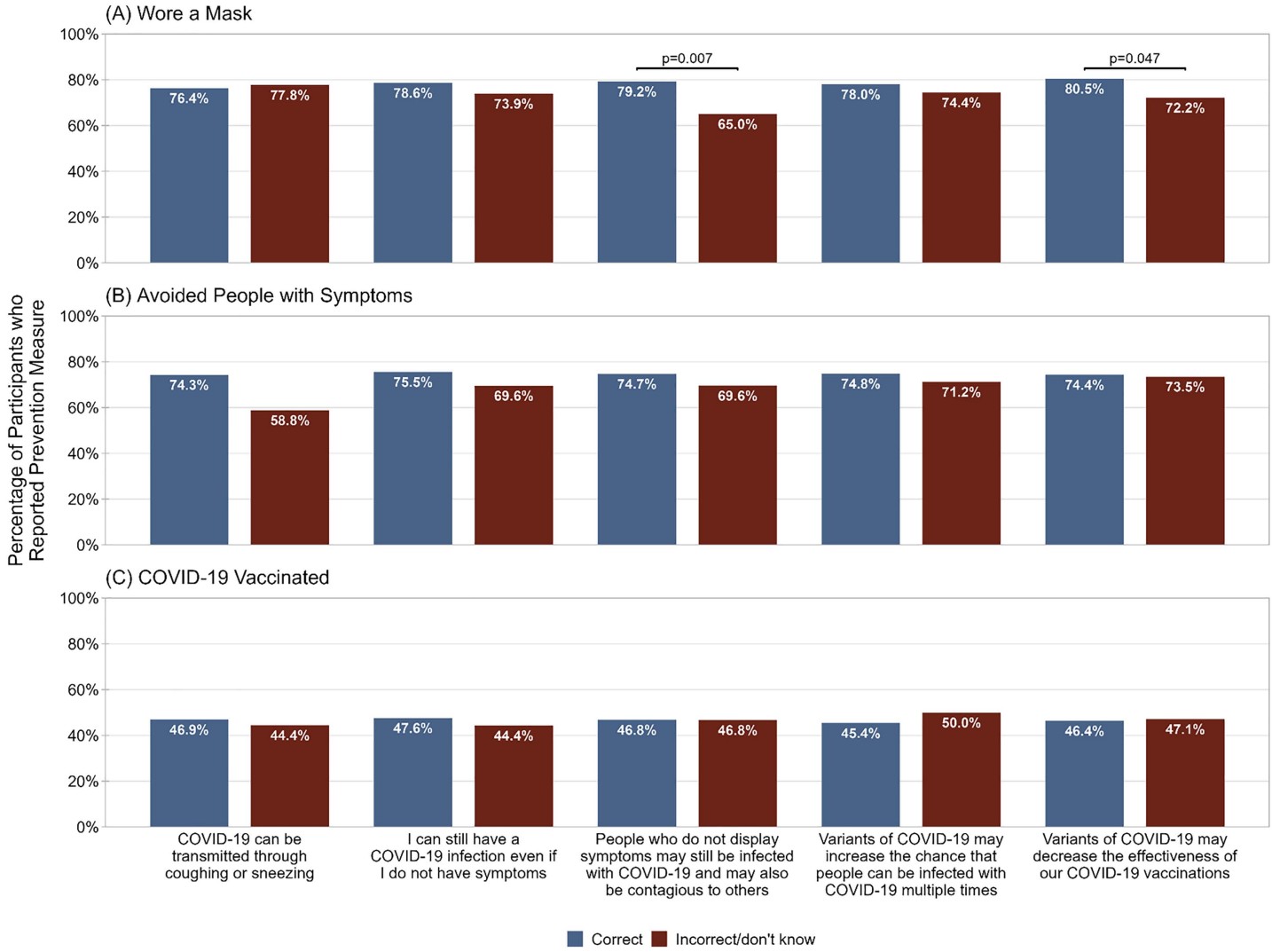

**Fig 2. Percentage of participants who reported A) wearing a mask in the past month, B) avoiding people with symptoms in the past month, and C) ever having received a COVID-19 vaccine, among participants who correctly or incorrectly identified true statements regarding COVID-19.** Preventive practice was ascertained by asking participants if they had used various practices in the past month to prevent COVID-19. COVID-19 knowledge was ascertained by asking participants to identify statements regarding COVID-19 as True or False.

p = 0.014; Table 4). Finally, participants with a weekly household income ≤25000 Ugandan shillings were less likely to be vaccinated against COVID-19 (aPR 0.69 [95% CI 0.48–1.00], p = 0.048; Table 4).

## Discussion

In these analyses of adult and adolescent participants with behavioral vulnerability to HIV in Uganda, we found substantial variability in COVID-19 knowledge, attitudes, and practices. As in other studies, knowledge levels were high overall but varied by aspect of COVID-19 physiology and transmission queried [8–10]. The greatest understanding was of the mode of SARS-CoV-2 transmission, with fewer participants correctly identifying transmission from asymptomatic individuals, and even fewer participants correctly identifying the impacts of SARS-CoV-2 genetic variants. A prior study of healthcare

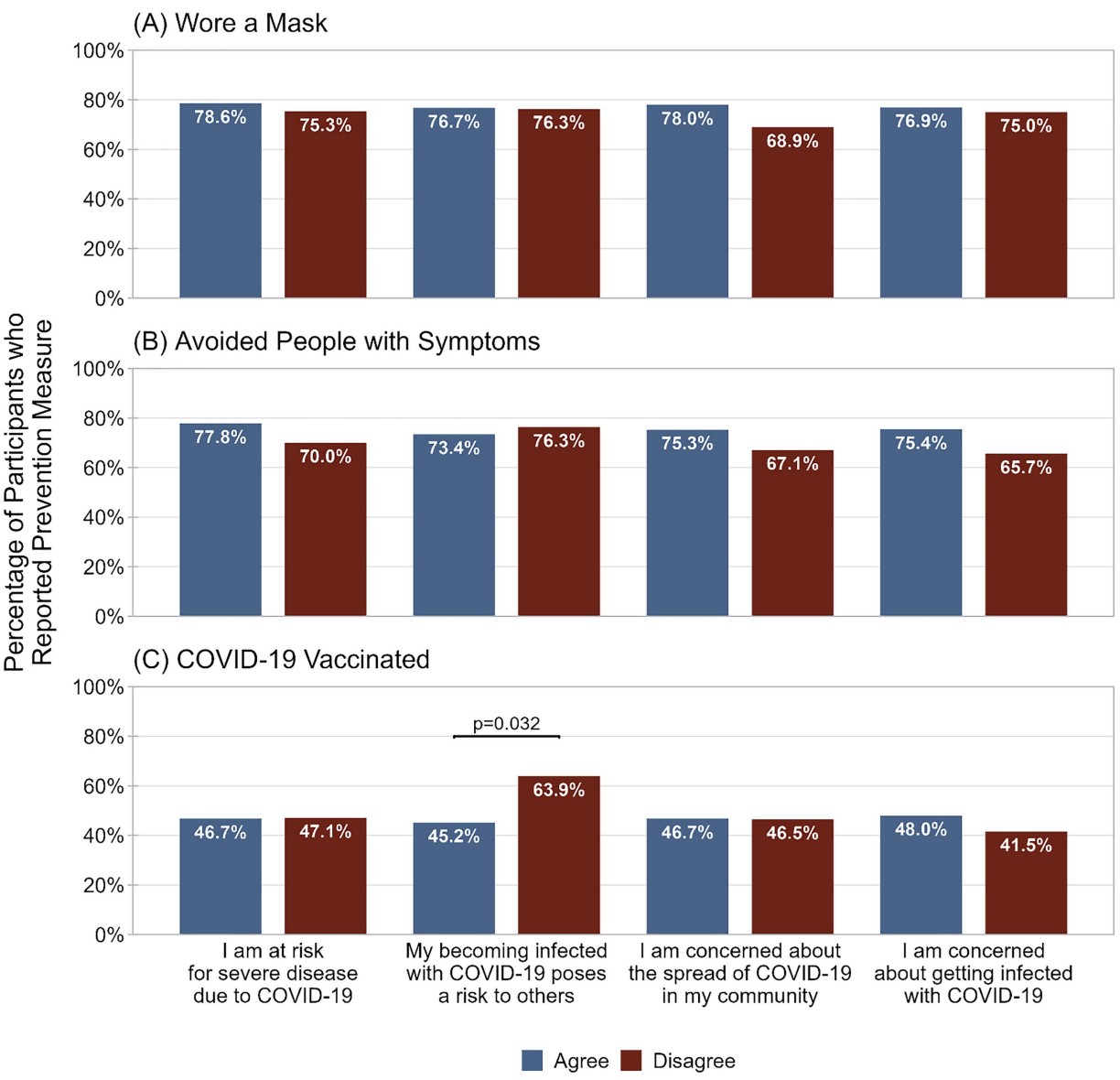

**Fig 3. Percentage of participants who reported A) wearing a mask in the last month, B) avoiding people with symptoms in the last month, and C) ever receiving a COVID-19 vaccine, among participants who agreed or disagreed with statements regarding COVID-19 risk.** Preventive practice was ascertained by asking participants if they had used various practices in the past month to prevent COVID-19. Attitudes to COVID-19 were ascertained by asking participants their level of agreement with statements regarding COVID-19 risk.

workers similarly found that knowledge of asymptomatic infection was the lowest among all aspects of COVID-19, despite that population being healthcare workers with 80% of participants completing secondary education compared to only 38% in our study suggesting that other factors such as public health messaging may be causing the low knowledge [10]. Attitudes toward COVID-19 also varied in this study, with generally cautious attitudes toward community transmission. However, fewer than half of participants believed they were at high risk of severe disease from COVID-19, potentially reflecting the young age of our participants. Finally, although most participants endorsed wearing facemasks and handwashing,

**Table 2. Sociodemographic characteristics, attitudes, and knowledge factors associated with wearing a face mask.**

| | Wearing a face mask | | |
|---|---|---|---|
| | PR (95% CI) | aPR (95% CI) | p-value |
| **Characteristic** | | | |
| Sex: | | | |
| Male | 1.00 (0.89-1.11) | | |
| Female | Ref | | |
| Age: | | | |
| 15–24 years | 0.92 (0.81-1.04) | | |
| 25–40 years | Ref | | |
| Education Level: | | | |
| <12 years | 0.92 (0.83-1.02) | 0.91 (0.81-1.01) | 0.068 |
| ≥12 years | Ref | Ref | |
| Weekly Household Income: | | | |
| ≤25000 Ugandan shillings | 1.05 (0.91-1.20) | | |
| >25000 Ugandan shillings | Ref | | |
| Knows Anyone with COVID-19: | | | |
| Yes | 1.00 (0.88-1.14) | | |
| No | Ref | | |
| COVID-19 Wave at Enrollment: | | | |
| BQ.1/BQ1.1 Omicron Wave | **0.82 (0.68-1.00)** | **0.81 (0.67-0.99)** | **0.034** |
| Non-Wave | Ref | Ref | |
| Sex Work or Transactional Sex: | | | |
| Yes | 1.13 (0.91-1.40) | | |
| No | Ref | | |
| **Attitude Statements** | | | |
| I am at risk for severe disease due to COVID-19: | | | |
| Agree | 1.05 (0.94-1.17) | | |
| Disagree/Neutral/Don't Know | Ref | | |
| My becoming infected with COVID-19 poses a risk to others: | | | |
| Agree | 1.01 (0.83-1.23) | | |
| Disagree/Neutral/Don't Know | Ref | | |
| I am concerned about the spread of COVID-19 in my community: | | | |
| Agree | 1.15 (0.97-1.37) | | |
| Disagree/Neutral/Don't Know | Ref | | |
| I am concerned about getting infected with COVID-19: | | | |
| Agree | 1.00 (0.86-1.16) | | |
| Disagree/Neutral/Don't Know | Ref | | |
| **Knowledge Statements** | | | |
| COVID-19 can be transmitted through coughing or sneezing | | | |
| Correct | 0.88 (0.72-1.07) | | |
| Incorrect | Ref | | |
| I can still have a COVID-19 infection even if I do not have symptoms | | | |
| Correct | 1.08 (0.96-1.22) | | |
| Incorrect | Ref | | |

*(Continued)*

**Table 2.** (Continued)

| | Wearing a face mask | | |
|---|---|---|---|
| | **PR (95% CI)** | **aPR (95% CI)** | **p-value** |
| People who do not display symptoms may still be infected with COVID-19 and may also be contagious to others | | | |
| Correct | **1.21 (1.01-1.44)** | 1.19 (1.00-1.43) | 0.052 |
| Incorrect | Ref | Ref | |
| Variants of COVID-19 may increase the chance that people can be infected with COVID-19 multiple times | | | |
| Correct | 1.07 (0.94-1.21) | | |
| Incorrect | Ref | | |
| Variants of COVID-19 may decrease the effectiveness of our COVID-19 vaccinations | | | |
| Correct | 1.10 (0.98-1.22) | | |
| Incorrect | Ref | | |

Variables in adjusted models were selected using purposeful variable selection. The number of participants included in this model was 380. **Bolded** values indicate variables significant at p<0.05.

PR: prevalence ratio; aPR: adjusted prevalence ratio; CI: confidence interval

fewer participants reported measures that decreased mobility such as reducing shopping trips, staying at home, and avoiding crowds, which has also been noted in prior studies [9].

Vaccination was also low in our study with only 47% of participants reporting a history of COVID-19 vaccination, although this level is higher than national estimates [33]. One prior cross-sectional study of COVID-19 vaccination in Uganda found lower COVID-19 vaccine willingness among people from urban locations during the early phases of vaccine roll-out [34]. Our study was conducted in an urban location but after the primary public health push for vaccination, likely accounting for the higher vaccination level. COVID-19 vaccinations in Uganda began around March 2021 for older individuals, and eligibility expanded to ≥18 years in August 2021, whereas our study began enrolling in October 2022. Although our study enrolled participants over one year after vaccination eligibility expanded, the proportion of participants vaccinated was low compared to other countries, consistent with prior studies showing plateauing vaccination levels [6]. Vaccination history was evaluated as a binary outcome in these analyses and did not consider vaccine type, timing, or number of doses, which could have informed a more nuanced understanding of this biomedical preventive practice. Finally, difficulty accessing vaccines has previously been reported which, along with WHO estimates that only half of the vaccines supplied to Uganda were ultimately administered and the barriers to healthcare faced by this study population, suggest that more efficient processes could have increased vaccination levels [34,35].

These analyses did not identify many predictors of COVID-19 preventive practices, suggesting that uptake of practices was uniform across most observed characteristics. Among the few predictors identified, however, were increased avoidance of symptomatic individuals and increased vaccination among male participants, contrasting with prior studies that have found that females were more like to endorse preventive practices [8]. This difference may at least partially be explained by our unique study population. For example, many participants were sex workers, for whom avoiding symptomatic people may not be feasible.

These analyses revealed the Omicron wave to be a unique period in the Ugandan COVID-19 pandemic. Participants were less likely to report wearing a face mask during the Omicron variant wave period, similar to a

**Table 3. Sociodemographic characteristics, attitudes, and knowledge factors associated with avoiding people with COVID-19 symptoms.**

| | Avoiding people with symptoms | | |
| --- | --- | --- | --- |
| | PR (95% CI) | aPR (95% CI) | p-value |
| **Characteristic** | | | |
| Sex: | | | |
| Male | **1.15 (1.02-1.30)** | **1.16 (1.03-1.31)** | **0.012** |
| Female | Ref | Ref | |
| Age: | | | |
| 15–24 years | 1.06 (0.90-1.26) | | |
| 25–40 years | Ref | | |
| Education Level: | | | |
| <12 years | 0.95 (0.84-1.07) | | |
| ≥12 years | Ref | | |
| Weekly Household Income: | | | |
| ≤25000 Ugandan shillings | 1.06 (0.92-1.23) | | |
| >25000 Ugandan shillings | Ref | | |
| Knows Anyone with COVID-19: | | | |
| Yes | 1.04 (0.90-1.20) | | |
| No | Ref | | |
| COVID-19 Wave at Enrollment: | | | |
| BQ.1/BQ1.1 Omicron Wave | 0.85 (0.69-1.03) | 0.84 (0.69-1.02) | 0.084 |
| Non-Wave | Ref | Ref | |
| Sex Work or Transactional Sex: | | | |
| Yes | 0.95 (0.79-1.14) | | |
| No | Ref | | |
| **Attitude Statements** | | | |
| I am at risk for severe disease due to COVID-19: | | | |
| Agree | 1.10 (0.98-1.24) | | |
| Disagree/Neutral/Don't Know | Ref | | |
| My becoming infected with COVID-19 poses a risk to others: | | | |
| Agree | 0.96 (0.79-1.17) | | |
| Disagree/Neutral/Don't Know | Ref | | |
| I am concerned about the spread of COVID-19 in my community: | | | |
| Agree | 1.15 (0.95-1.38) | 1.10 (0.89-1.36) | 0.364 |
| Disagree/Neutral/Don't Know | Ref | Ref | |
| I am concerned about getting infected with COVID-19: | | | |
| Agree | 1.15 (0.94-1.40) | 1.10 (0.88-1.37) | 0.394 |
| Disagree/Neutral/Don't Know | Ref | Ref | |
| **Knowledge Statements** | | | |
| COVID-19 can be transmitted through coughing or sneezing | | | |
| Correct | 1.32 (0.85-2.05) | | |
| Incorrect | Ref | | |
| I can still have a COVID-19 infection even if I do not have symptoms | | | |
| Correct | 1.09 (0.95-1.25) | | |
| Incorrect | Ref | | |
| People who do not display symptoms may still be infected with COVID-19 and may also be contagious to others | | | |
| Correct | 1.05 (0.89-1.24) | | |
| Incorrect | Ref | | |

*(Continued)*

**Table 3.** (Continued)

| | Avoiding people with symptoms | | |
| --- | --- | --- | --- |
| | PR (95% CI) | aPR (95% CI) | p-value |
| Variants of COVID-19 may increase the chance that people can be infected with COVID-19 multiple times | | | |
| Correct | 1.05 (0.91-1.20) | | |
| Incorrect | Ref | | |
| Variants of COVID-19 may decrease the effectiveness of our COVID-19 vaccinations | | | |
| Correct | 0.98 (0.87-1.11) | | |
| Incorrect | Ref | | |

Variables in adjusted models were selected using purposeful variable selection. The number of participants included in this model was 379. **Bolded** values indicate variables significant at p < 0.05.

PR: prevalence ratio; aPR: adjusted prevalence ratio; CI: confidence interval

prior study finding decreased NPI practice coinciding with increased SARS-CoV-2 transmission [36]. This is also consistent with prior data from Kenya showing decreased self-report of multiple COVID-19 preventive practices during the Omicron wave, and studies generally demonstrating varying use of preventive practices over time [37–39]. Finally, the analyses also identified increased COVID-19 vaccination during the Omicron variant wave despite the Omicron period (31 October 2022−2 January 2023) consisting of only two months early in the one year of total enrollment. One possible explanation is that COVID-19-vaccinated individuals may be more likely to enroll in this study during periods of high community SARS-CoV-2 transmission than unvaccinated individuals. Another possibility is that earlier enrollees may have been more recently vaccinated and therefore more likely to recall vaccination when queried.

These analyses provided valuable data to inform preventive interventions during future public health crises. However, there were several limitations. First, our period of enrollment began in October 2022, well after vaccination campaigns had ramped up and public health perceptions may have evolved as compared to earlier in the pandemic, when prior studies of COVID-19 knowledge, attitudes, and practices in Uganda had been performed. Second, the data used were self-reported and may be susceptible to biases, including recall and social desirability biases. However, computer-assisted self-interviews were used when possible to mitigate social desirability bias and participants were asked to report NPI practices specifically from only the previous month, which helped minimize recall bias. Third, assessments of knowledge, attitudes, and practices are not standardized across the literature which limits comparability, but they generally have similar themes despite varying wording. Lastly, the MOCHI study specifically enrolls participants with behavioral vulnerabilities to HIV and consequently, is not representative of the general population. For example, our study population was predominantly adolescents and young adults, and most males were MSM. However, these individuals face particularly high barriers to healthcare given the overlapping risk factors for HIV and COVID-19, and understanding their knowledge, attitudes, and practices may provide outsized benefits in future pandemics. [40]

These analyses identified incomplete utilization of COVID-19 NPIs and vaccination among individuals with behavioral vulnerabilities to HIV. Increasing knowledge of nuanced aspects of disease outbreaks—such as the possibility for asymptomatic transmission and the effects of genetic variants—may increase uptake and adherence to preventive practices in future respiratory pandemics.

**Table 4.** Sociodemographic characteristics, attitudes, and knowledge factors associated with COVID-19 vaccination.

| | COVID-19 vaccination | | |
| --- | --- | --- | --- |
| | PR (95% CI) | aPR (95% CI) | p-value |
| **Characteristic** | | | |
| Sex: | | | |
| Male | **1.30 (1.05-1.61)** | **1.30 (1.05-1.60)** | **0.014** |
| Female | Ref | Ref | |
| Age: | | | |
| 15–24 years | 0.83 (0.64-1.06 | | |
| 25–40 years | Ref | | |
| Education Level: | | | |
| < 12 years | 0.82 (0.66-1.02) | | |
| ≥ 12 years | Ref | | |
| Weekly Household Income: | | | |
| ≤ 25000 Ugandan shillings | **0.65 (0.45-0.94)** | **0.69 (0.48-1.00)** | **0.048** |
| > 25000 Ugandan shillings | Ref | Ref | |
| Knows Anyone with COVID-19: | | | |
| Yes | 0.83 (0.62-1.12) | | |
| No | Ref | | |
| COVID-19 Wave at Enrollment: | | | |
| BQ.1/BQ1.1 Omicron Wave | **1.68 (1.36-2.07)** | **1.59 (1.29-1.95)** | **<0.001** |
| Non-Wave | Ref | Ref | |
| Sex Work or Transactional Sex: | | | |
| Yes | 0.90 (0.65-1.25) | | |
| No | Ref | | |
| **Attitude Statements** | | | |
| I am at risk for severe disease due to COVID-19: | | | |
| Agree | 0.99 (0.8-1.23) | | |
| Disagree/Neutral/Don't Know | Ref | | |
| My becoming infected with COVID-19 poses a risk to others: | | | |
| Agree | 0.75 (0.55-1.01) | 0.76 (0.56-1.02) | 0.071 |
| Disagree/Neutral/Don't Know | Ref | Ref | |
| I am concerned about the spread of COVID-19 in my community: | | | |
| Agree | 0.94 (0.71-1.24) | | |
| Disagree/Neutral/Don't Know | Ref | | |
| I am concerned about getting infected with COVID-19: | | | |
| Agree | 1.08 (0.78-1.47) | | |
| Disagree/Neutral/Don't Know | Ref | | |
| **Knowledge Statements** | | | |
| COVID-19 can be transmitted through coughing or sneezing | | | |
| Correct | 1.07 (0.61-1.88) | | |
| Incorrect | Ref | | |
| I can still have a COVID-19 infection even if I do not have symptoms | | | |
| Correct | 1.11 (0.87-1.40) | | |
| Incorrect | Ref | | |

*(Continued)*

**Table 4.** (Continued)

| | COVID-19 vaccination | | |
| --- | --- | --- | --- |
| | **PR (95% CI)** | **aPR (95% CI)** | **p-value** |
| People who do not display symptoms may still be infected with COVID-19 and may also be contagious to others | | | |
| Correct | 1.01 (0.76-1.35) | | |
| Incorrect | Ref | | |
| Variants of COVID-19 may increase the chance that people can be infected with COVID-19 multiple times | | | |
| Correct | 0.92 (0.73-1.16) | | |
| Incorrect | Ref | | |
| Variants of COVID-19 may decrease the effectiveness of our COVID-19 vaccinations | | | |
| Correct | 0.99 (0.80-1.23) | | |
| Incorrect | Ref | | |

Variables in adjusted models were selected using purposeful variable selection. The number of participants included in this model was 373. **Bolded** values indicate variables significant at p<0.05.

PR: prevalence ratio; aPR: adjusted prevalence ratio; CI: confidence interval

## Supporting information

**S1 File. Checklist.**
(DOCX)

## Acknowledgments

We would like to thank the MOCHI participants and the members of the study team for their contributions. MOCHI Study Group:

Henry M. Jackson Foundation for the Advancement of Military Medicine, Bethesda, USA: Julius Tonzel, Roger Ying, Curtisha Charles, Linsey Scheibler, Tsedal Mebrahtu, Brian Liles, Bryce Boron, Ying Fan, Qun Li, Alexus Reynolds, Glenna Schluck, Natalie Burns, Leigh Anne Eller, Michelle Imbach, Jacob Peterson, Addison Walling, Haoyu Qian, and Trevor Crowell (protocol chair and lead author, tcrowell@hivresearch.org)

U.S. Military HIV Research Program, CIDR, Walter Reed Army Institute of Research, Silver Spring, Maryland, USA: Julie Ake, Paul Adjei, Brennan Cebula, Julius Tonzel, Roger Ying, Curtisha Charles, Linsey Scheibler, Tsedal Mebrahtu, Brian Liles, Bryce Boron, Ying Fan, Qun Li, Alexus Reynolds, Glenna Schluck, Natalie Burns, Leigh Anne Eller, Michelle Imbach, Jacob Peterson, Addison Walling, Haoyu Qian, and Trevor Crowell (protocol chair and lead author, tcrowell@hivresearch.org)

Makerere University-Walter Reed Program, Makerere University, Kampala, Uganda: Betty Mwesigwa (Uganda principal investigator), Hannah Kibuuka, Grace Mirembe, Immaculate Nakabuye, Prossy Naluyima, Ronald Ephraim Wasswa, Justine Nalunga and Emmanuel Wasswa

U.S. Military HIV Research Program, Walter Reed Army Institute of Research – Africa, Kampala, Uganda: Betty Mwesigwa (Uganda principal investigator), Hannah Kibuuka, Grace Mirembe, Immaculate Nakabuye, Prossy Naluyima, Ronald Ephraim Wasswa, Justine Nalunga and Emmanuel Wasswa

## Disclaimer

This material has been reviewed by the Walter Reed Army Institute of Research. There is no objection to its presentation and/or publication. The opinions or assertions contained herein are the private views of the author, and are not to be construed as official, or as reflecting true views of the Department of the Army, the Department of Defense, or HJF. The investigators have adhered to the policies for protection of human participants as prescribed in AR 70–25.

Prior presentation: This work was previously presented, in part, at the 13th IAS Conference on HIV Science in Kigali, Rwanda, 13–17 July 2025.

## Author contributions

**Conceptualization:** Job Kasule, Trevor A. Crowell.

**Data curation:** Natalie Burns, Tyler Hamby.

**Formal analysis:** Natalie Burns, Tyler Hamby.

**Funding acquisition:** Trevor A. Crowell.

**Methodology:** Julius L. Tonzel, Roger Ying, Margaret Yacovone, Trevor A. Crowell.

**Project administration:** Grace Mirembe, Immaculate Nakabuye, Hannah Kibuuka, Betty Mwesigwa.

**Supervision:** Margaret Yacovone.

**Writing – original draft:** Job Kasule.

**Writing – review & editing:** Julius L. Tonzel, Roger Ying, Betty Mwesigwa, Trevor A. Crowell.

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
