## [Decision Letter · Decision Letter 0]

15 Dec 2025

PONE-D-25-48971COVID-19 knowledge, attitudes, and practices among people vulnerable to HIV in Uganda: A cross-sectional cohort analysisPLOS One

Dear Dr. Ying,

Thank you for submitting your manuscript to PLOS ONE. After careful consideration, we feel that it has merit but does not fully meet PLOS ONE’s publication criteria as it currently stands. Therefore, we invite you to submit a revised version of the manuscript that addresses the points raised during the review process.

We look forward to receiving your revised manuscript.

Kind regards,

Armaan Jamal

Guest Editor

PLOS One

Journal Requirements:

When submitting your revision, we need you to address these additional requirements

“This work was supported by agreements between the Henry M. Jackson Foundation for the Advancement of Military Medicine, Inc., and the U.S. Department of Defense (W81XWH-18-2-0040; HT9425-24-3-0004). This research was funded, in part, by the U.S. National Institute of Allergy and Infectious Diseases (AAI20052001) and National Institute of Mental Health (RF1 MH133442: Crowell)”

“This work was supported by agreements between the Henry M. Jackson Foundation for the Advancement of Military Medicine, Inc., and the U.S. Department of Defense (W81XWH-18-2-0040; HT9425-24-3-0004). This research was funded, in part, by the U.S. National Institute of Allergy and Infectious Diseases (AAI20052001) and National Institute of Mental Health (RF1 MH133442: Crowell)”

5. We note that you have indicated that there are restrictions to data sharing for this study. For studies involving human research participant data or other sensitive data, we encourage authors to share de-identified or anonymized data. However, when data cannot be publicly shared for ethical reasons, we allow authors to make their data sets available upon request. For information on unacceptable data access restrictions, please see http://journals.plos.org/plosone/s/data-availability#loc-unacceptable-data-access-restrictions.

6. In this instance it seems there may be acceptable restrictions in place that prevent the public sharing of your minimal data. However, in line with our goal of ensuring long-term data availability to all interested researchers, PLOS’ Data Policy states that authors cannot be the sole named individuals responsible for ensuring data access (http://journals.plos.org/plosone/s/data-availability#loc-acceptable-data-sharing-methods).

7. Please be informed that funding information should not appear in the Acknowledgments section or other areas of your manuscript. We will only publish funding information present in the Funding Statement section of the online submission form. Please remove any funding-related text from the manuscript.

8. One of the noted authors is a group or consortium [Multinational Observational Cohort of HIV and other Infections (MOCHI) Study Group]. In addition to naming the author group, please list the individual authors and affiliations within this group in the acknowledgments section of your manuscript. Please also indicate clearly a lead author for this group along with a contact email address.

Reviewers' comments:

Reviewer's Responses to Questions

**Comments to the Author**

1. Is the manuscript technically sound, and do the data support the conclusions?

Reviewer #1: Yes

Reviewer #2: Partly

Reviewer #3: Yes

2. Has the statistical analysis been performed appropriately and rigorously? 

Reviewer #1: Yes

Reviewer #2: Yes

Reviewer #3: I Don't Know

3. Have the authors made all data underlying the findings in their manuscript fully available?

Reviewer #1: Yes

Reviewer #2: Yes

Reviewer #3: Yes

4. Is the manuscript presented in an intelligible fashion and written in standard English?

Reviewer #1: Yes

Reviewer #2: No

Reviewer #3: Yes

5. Review Comments to the Author

Reviewer #1: This study presents a technically sound and well-written analysis, demonstrating the variable COVID-19 knowledge and attitudes among Ugandan adolescents and adults with little impact on COVID-19 preventive practices. It is an exciting article to publish. The title is COVID-19 knowledge, attitudes, and practices among people vulnerable to HIV in Uganda: A cross-sectional cohort analysis.

Reviewer #2: This manuscript presents a cross-sectional analysis of COVID-19 knowledge, attitudes, and practices among individuals with vulnerability to HIV in Uganda and addresses an important hard-to-reach population; however, several major limitations should be addressed:

1. The study population is highly specific (predominantly young, 15-24 years old (81%)), which severely limits external validity, yet the conclusions are framed broadly for future pandemics.

2. Vaccination is analyzed only as “ever vaccinated” without consideration of dose number, timing, or booster status, substantially weakening vaccine-related conclusions. The authors should give details if data is available.

3. The use of complete-case analysis without reporting the extent of missing data or conducting sensitivity analyses raises concerns about selection bias. I think you could perform sensitivity analyses.

4. The authors used both terms, COVID-19 and SARS-CoV-2, which is inconsistent; please select one of them and be consistent.

5. Table 2 is dense and difficult to interpret. Would you be able to make it clearer?

6. The other point is the inclusion of 15–17-year-olds without parental consent, although it seems legally justified locally, remains highly sensitive internationally. You could provide some explanation somewhere in your manuscript.

Reviewer #3: General Comment

The study aims to describe the knowledge, attitudes, and practices regarding public health measures to reduce COVID-19 transmission among adults and adolescents engaged in HIV-susceptible behaviour. While the topic is relevant, several methodological and ethical issues limit the overall value and generalizability of the findings.

Major Concerns

1. Overly Specific Study Population

The study population is extremely specific, which significantly limits the generalizability of the findings. Although the authors acknowledge this limitation, the issue remains substantial and reduces the broader applicability of the results.

2. Vulnerable Participants and Ethical Considerations

Some participants are minors and are also described as having risky behaviours (e.g., alcohol use, drug dependence, alcohol dependence, or a history of STIs). The manuscript does not clarify how these minors were deemed mature enough to provide informed consent. Ethical standards require that informed consent for minors be obtained from parents or legal guardians. This aspect needs to be clearly addressed and justified.

Minor Concerns

1. Definition of Non-Pharmaceutical Interventions

The authors should clearly define what they mean by “non-pharmaceutical interventions” in the context of COVID-19 prevention. Additionally, clarification is needed regarding whether pharmaceutical interventions (e.g., antiviral treatments) were relevant or available during the study period.

2. Context of Vaccination and Socioeconomic Factors

The manuscript reports associations between vaccination status, income levels, and the waves of COVID-19. However, the authors do not provide adequate context about the vaccination program in Uganda at the time. For instance, were vaccines provided free of charge to the public? Such information is essential for interpreting the findings.

3. Self-Interview Bias

The use of self-interviews introduces potential bias. Although this is acknowledged in the limitations section, the authors do not explain how such bias was minimized or managed, which should be addressed in the methodology.

4. Exclusion of Pregnant Women

There is no explanation for the exclusion of pregnant women from the study. The authors should justify this criterion, especially if the exclusion may affect the representativeness of the sample.

6. PLOS authors have the option to publish the peer review history of their article (what does this mean? ). If published, this will include your full peer review and any attached files.

**Do you want your identity to be public for this peer review?** For information about this choice, including consent withdrawal, please see our Privacy Policy .

Reviewer #1: No

Reviewer #2: **Yes:** Dr Omeed Darweesh

Reviewer #3: No

---

## [Author Response · Author response to Decision Letter 1]

13 Jan 2026

COMMENTS

Reviewer #1

This study presents a technically sound and well-written analysis, demonstrating the variable COVID-19 knowledge and attitudes among Ugandan adolescents and adults with little impact on COVID-19 preventive practices. It is an exciting article to publish. The title is COVID-19 knowledge, attitudes, and practices among people vulnerable to HIV in Uganda: A cross-sectional cohort analysis.

Response: The authors appreciate the complimentary comments from the reviewer.

Reviewer #2

This manuscript presents a cross-sectional analysis of COVID-19 knowledge, attitudes, and practices among individuals with vulnerability to HIV in Uganda and addresses an important hard-to-reach population; however, several major limitations should be addressed:

1. The study population is highly specific (predominantly young, 15-24 years old (81%)), which severely limits external validity, yet the conclusions are framed broadly for future pandemics.

Response: Thank you for this observation. We agree that generalizability is limited, and note that these participants represent people with large barriers to healthcare. We have edited the Discussion to further emphasize the limited generalizability: “Lastly, the MOCHI study specifically enrolls participants with behavioral vulnerabilities to HIV and consequently, is not representative of the general population. For example, our study population is predominantly adolescents and young adults, and most males are MSM. However, these individuals face particularly high barriers to healthcare given the overlapping risk factors for HIV and COVID-19, and understanding their knowledge, attitudes, and practices may provide outsized benefits in future pandemics [40].” (Lines 328-331)

40. Auerbach JD, Forsyth AD, Davey C, Hargreaves JR, Group for lessons from pandemic HIVpftC-r. Living with COVID-19 and preparing for future pandemics: revisiting lessons from the HIV pandemic. Lancet HIV. 2023;10(1):e62-e8. Epub 20221110. doi: 10.1016/S2352-3018(22)00301-0. PubMed PMID: 36370713; PubMed Central PMCID: PMCPMC9764384.

2. Vaccination is analyzed only as “ever vaccinated” without consideration of dose number, timing, or booster status, substantially weakening vaccine-related conclusions. The authors should give details if data is available.

Response: We agree with the reviewer that those aspects of vaccination history would be interesting to evaluate in order to draw vaccine-related considerations. We were not able to evaluate these and have edited the text to reflect this limitation: “Although our study enrolled participants over one year after vaccination eligibility expanded, the proportion of participants vaccinated was low compared to other countries, consistent with prior studies showing plateauing vaccination levels [6]. Vaccination history was evaluated as a binary outcome in these analyses and did not consider vaccine type, timing, or number of doses, which could have informed a more nuanced understanding of this biomedical preventive practice.” (Lines 288-293)

6. King P, Wanyana MW, Migisha R, Kadobera D, Kwesiga B, Claire B, et al. COVID-19 Vaccine Uptake and Coverage, Uganda, 2021-2022. Uganda National Institute of Public Health Quarterly Epidemiological Bulletin. 2023;8(1).

3. The use of complete-case analysis without reporting the extent of missing data or conducting sensitivity analyses raises concerns about selection bias. I think you could perform sensitivity analyses.

Response: Thank you for raising this concern. The proportion of missing data was consistent for all through multivariable models at approximately 9%. We have included these percentages in the text: “In multivariable analyses, most participants had complete data to be included in multivariable modeling (380/417 [91.1%] for wearing a face mask, 379/415 [91.3%] for avoiding people with symptoms, and 373/409 [91.2%] for receiving a COVID-19 vaccine).” (Lines 235-237)

4. The authors used both terms, COVID-19 and SARS-CoV-2, which is inconsistent; please select one of them and be consistent.

Response: Thank you for this observation. We aimed to use SARS-CoV-2 to refer to the virus (e.g. “SARS-CoV-2 transmission”) and COVID-19 to refer to the disease caused by SARS-CoV-2 (e.g. “severe disease from COVID-19”). We have revised the manuscript to ensure consistency throughout. However, where the questionnaire is quoted directly, COVID-19 is sometimes used to refer to the virus. The questionnaire was drafted early in the pandemic and did not reflect the current standards for terminology. For transparency, the authors have retained the exact questions, prompts, and response options that were provided to participants without updating terminology to the current standard.

5. Table 2 is dense and difficult to interpret. Would you be able to make it clearer?

Response: That you for this comment. We have divided Table 2 into three tables, one for each outcome model (wearing a face mask, avoiding people with symptoms, and COVID-19 vaccination) (Tables 2-4)

6. The other point is the inclusion of 15–17-year-olds without parental consent, although it seems legally justified locally, remains highly sensitive internationally. You could provide some explanation somewhere in your manuscript.

Response: Thank you for this comment. We included adolescents because of their known high STI incidence, and per local guidelines, participants aged 14-17 with a history of STIs were considered mature adolescents who could consent to the study on their own. We have clarified these points in the text.

“Participants as young as 14 years were considered eligible for inclusion because of known early sexual debut in Uganda and East Africa, which has been associated with high prevalence and incidence of HIV and other STIs [28-32]. According to local guidelines, participants aged 14-17 years with drug dependency, alcohol dependency, or a history STI were considered mature; participants aged 14-17 years who were pregnant, married, had a child, or catered for their own livelihood were considered emancipated. Consent was not sought from parents or guardians of mature or emancipated minors. Minors who were not considered mature or emancipated were not enrolled.” (Lines 137-144)

28. Grabowski MK, Mpagazi J, Kiboneka S, Ssekubugu R, Kereba JB, Nakayijja A, et al. The HIV and sexually transmitted infection syndemic following mass scale-up of combination HIV interventions in two communities in southern Uganda: a population-based cross-sectional study. Lancet Glob Health. 2022;10(12):e1825-e34. doi: 10.1016/S2214-109X(22)00424-7. PubMed PMID: 36400088; PubMed Central PMCID: PMCPMC10068679.

29. Masanja V, Wafula ST, Ssekamatte T, Isunju JB, Mugambe RK, Van Hal G. Trends and correlates of sexually transmitted infections among sexually active Ugandan female youths: evidence from three demographic and health surveys, 2006-2016. BMC Infect Dis. 2021;21(1):59. Epub 20210113. doi: 10.1186/s12879-020-05732-x. PubMed PMID: 33435882; PubMed Central PMCID: PMCPMC7805221.

30. Sing'oei V, Owuoth JK, Otieno J, Yates A, Andagalu B, Smith HJ, et al. Early sexual debut is associated with drug use and decreased educational attainment among males and females in Kisumu County, Kenya. Reprod Health. 2023;20(1):111. Epub 20230727. doi: 10.1186/s12978-023-01639-3. PubMed PMID: 37501066; PubMed Central PMCID: PMCPMC10375697.

31. Awuoche HC, Joseph RH, Magut F, Khagayi S, Odongo FS, Otieno M, et al. Prevalence and risk factors of sexually transmitted infections in the setting of a generalized HIV epidemic-a population-based study, western Kenya. Int J STD AIDS. 2024;35(6):418-29. Epub 20240119. doi: 10.1177/09564624241226487. PubMed PMID: 38240604; PubMed Central PMCID: PMCPMC11047016.

32. Mudhune V, Winskell K, Bednarczyk RA, Ondenge K, Mbeda C, Kerubo E, et al. Sexual behaviour among Kenyan adolescents enrolled in an efficacy trial of a smartphone game to prevent HIV: a cross-sectional analysis of baseline data. SAHARA J. 2024;21(1):2320188. Epub 20240222. doi: 10.1080/17290376.2024.2320188. PubMed PMID: 38388022; PubMed Central PMCID: PMCPMC10885754.

Reviewer #3

The study aims to describe the knowledge, attitudes, and practices regarding public health measures to reduce COVID-19 transmission among adults and adolescents engaged in HIV-susceptible behaviour. While the topic is relevant, several methodological and ethical issues limit the overall value and generalizability of the findings.

Major Concerns

7. Overly Specific Study Population

The study population is extremely specific, which significantly limits the generalizability of the findings. Although the authors acknowledge this limitation, the issue remains substantial and reduces the broader applicability of the results.

Response: Thank you for this comment. We agree that the study population is not representative of the general population and is instead designed to represent a population with high HIV/STI incidence and that may be recruited for future trials of HIV prevention products. We have further clarified this limitation, and also identified how the specific study population may explain our unique findings.

“Among the few predictors identified, however, were increased avoidance of symptomatic individuals and increased vaccination among male participants, contrasting with prior studies that have found that females were more like to endorse preventive practices [8]. This difference may at least partially be explained by our unique study population. For example, many participants were sex workers, for whom avoiding symptomatic people may not be feasible.” (Lines 298-303)

8. Okello G, Izudi J, Teguzirigwa S, Kakinda A, Van Hal G. Findings of a Cross-Sectional Survey on Knowledge, Attitudes, and Practices about COVID-19 in Uganda: Implications for Public Health Prevention and Control Measures. Biomed Res Int. 2020;2020:5917378. Epub 20201204. doi: 10.1155/2020/5917378. PubMed PMID: 34031643; PubMed Central PMCID: PMCPMC7729389.

8. Vulnerable Participants and Ethical Considerations

Some participants are minors and are also described as having risky behaviours (e.g., alcohol use, drug dependence, alcohol dependence, or a history of STIs). The manuscript does not clarify how these minors were deemed mature enough to provide informed consent. Ethical standards require that informed consent for minors be obtained from parents or legal guardians. This aspect needs to be clearly addressed and justified.

Response: Thank you for this comment. We have clarified as follows:

“Participants as young as 14 years were considered eligible for inclusion because of known early sexual debut in Uganda and East Africa, which has been associated with high prevalence and incidence of HIV and other STIs [28-32]. According to local guidelines, participants aged 14-17 years with drug dependency, alcohol dependency, or a history STI were considered mature; participants aged 14-17 years who were pregnant, married, had a child, or catered for their own livelihood were considered emancipated. Consent was not sought from parents or guardians of mature or emancipated minors. Minors who were not considered mature or emancipated were not enrolled.” (Lines 137-144)

28. Grabowski MK, Mpagazi J, Kiboneka S, Ssekubugu R, Kereba JB, Nakayijja A, et al. The HIV and sexually transmitted infection syndemic following mass scale-up of combination HIV interventions in two communities in southern Uganda: a population-based cross-sectional study. Lancet Glob Health. 2022;10(12):e1825-e34. doi: 10.1016/S2214-109X(22)00424-7. PubMed PMID: 36400088; PubMed Central PMCID: PMCPMC10068679.

29. Masanja V, Wafula ST, Ssekamatte T, Isunju JB, Mugambe RK, Van Hal G. Trends and correlates of sexually transmitted infections among sexually active Ugandan female youths: evidence from three demographic and health surveys, 2006-2016. BMC Infect Dis. 2021;21(1):59. Epub 20210113. doi: 10.1186/s12879-020-05732-x. PubMed PMID: 33435882; PubMed Central PMCID: PMCPMC7805221.

30. Sing'oei V, Owuoth JK, Otieno J, Yates A, Andagalu B, Smith HJ, et al. Early sexual debut is associated with drug use and decreased educational attainment among males and females in Kisumu County, Kenya. Reprod Health. 2023;20(1):111. Epub 20230727. doi: 10.1186/s12978-023-01639-3. PubMed PMID: 37501066; PubMed Central PMCID: PMCPMC10375697.

31. Awuoche HC, Joseph RH, Magut F, Khagayi S, Odongo FS, Otieno M, et al. Prevalence and risk factors of sexually transmitted infections in the setting of a generalized HIV epidemic-a population-based study, western Kenya. Int J STD AIDS. 2024;35(6):418-29. Epub 20240119. doi: 10.1177/09564624241226487. PubMed PMID: 38240604; PubMed Central PMCID: PMCPMC11047016.

32. Mudhune V, Winskell K, Bednarczyk RA, Ondenge K, Mbeda C, Kerubo E, et al. Sexual behaviour among Kenyan adolescents enrolled in an efficacy trial of a smartphone game to prevent HIV: a cross-sectional analysis of baseline data. SAHARA J. 2024;21(1):2320188. Epub 20240222. doi: 10.1080/17290376.2024.2320188. PubMed PMID: 38388022; PubMed Central PMCID: PMCPMC10885754.

Minor Concerns

9. Definition of Non-Pharmaceutical Interventions

The authors should clearly define what they mean by “non-pharmaceutical interventions” in the context of COVID-19 prevention. Additionally, clarification is needed regarding whether pharmaceutical interventions (e.g., antiviral treatments) were relevant or available during the study period.

Response: Thank you for this comment. We have clarified the meaning of “non-pharmaceutical interventions” with the following:

“Early in the pandemic, Uganda’s government mandated non-pharmaceutical interventions (NPIs) to reduce SARS-CoV-2 transmission. Broadly, NPIs are strategies other than biomedical interventions that an individual can utilize to minimize their risk of infection, such as wearing face masks in public, social distancing, and avoiding social gatherings [2-4].” (Lines 29-33)

2. Squarcina M, Carraro A. Changing profiles of child poverty: The case of Uganda during the COVID-19 pandemic. UNICEF Innocenti: 2024.

3. Liu Y, Wang W, Wong WK, Zhu W. Effectiveness of non-pharmaceutical interventions for COVID-19 in USA. Sci Rep. 2024;14(1):21387. Epub 20240913. doi: 10.1038/s41598-024-71984-1. PubMed PMID: 39271786; PubMed Central PMCID: PMCPMC11399256.

4. Flaxman S, Mishra S, Gandy A, Unwin HJT, Mellan TA, Coupland H, et al. Estimating the effects of non-pharmaceutical interventions on COVID-19 in Europe. Nature. 2020;584(7820):257-61. Epub 20200608. doi: 10.1038/s41586-020-2405-7. PubMed PMID: 32512579.

Furthermore, non-vaccine pharmaceutical interventions such as antivirals were not widely available during the study period which is now reflected in the text:

“As COVID-19 pharmaceutical interventions were developed, NPI mandates were reduced [11]. However, therapeutic interventions such as antivirals (e.g., remdesivir) and monoclonal antibodies were not part of routine clinical care, which primarily relied on supportive interventions [12]. Therefore vaccination, beginning in March 2021, became the mainstay biomedical intervention [13].” (Lines 45-48)

11. Laing N, Mylan S, Parker M. Does epidemiological evidence support the success story of Uganda's response to COVID-19? J Biosoc Sci. 2024:1-8. Epub 20240311. doi: 10.1017/S0021932024000117. PubMed PMID: 38462976; PubMed Central PMCID: PMCPMC7616485.

12. Bongomin F, Fleischer B, Olum R, Natukunda B, Kiguli S, Byakika-Kibwika P, et al. High Mortality During the Second Wave of the Coronavirus Disease 2019 (COVID-19) Pandemic in Uganda: Experience From a National Referral COVID-19 Treatment Unit. Open Forum Infect Dis. 2021;8(11):ofab530. Epub 20211118. doi: 10.1093/ofid/ofab530. PubMed PMID: 34805440; PubMed Central PMCID: PMCPMC8601041.

13. Kiiza D, Semanda JN, Kawere BB, Ajore C, Wasswa CK, Kwiringira A, et al. Strategies to Enhance COVID-19 Vaccine Uptake among Prioritized Groups, Uganda-Lessons Learned and Recommendations for Future Pandemics. Emerg Infect Dis. 2024;30(7):1326-34. doi: 10.3201/eid3007.231001. PubMed PMID: 38916545; PubMed Central PMCID: PMCPMC11210662.

10. Context of Vaccination and Socioeconomic Factors

The manuscript reports associations between vaccination status, income levels, and t

---

## [Decision Letter · Decision Letter 1]

8 Feb 2026

COVID-19 knowledge, attitudes, and practices among people vulnerable to HIV in Uganda: A cross-sectional cohort analysis

PONE-D-25-48971R1

Dear Dr. Ying,

We’re pleased to inform you that your manuscript has been judged scientifically suitable for publication and will be formally accepted for publication once it meets all outstanding technical requirements.

Kind regards,

Armaan Jamal

Guest Editor

PLOS One

Reviewers' comments:

Reviewer's Responses to Questions

**Comments to the Author**

1. If the authors have adequately addressed your comments raised in a previous round of review and you feel that this manuscript is now acceptable for publication, you may indicate that here to bypass the “Comments to the Author” section, enter your conflict of interest statement in the “Confidential to Editor” section, and submit your "Accept" recommendation.

Reviewer #1: All comments have been addressed

2. Is the manuscript technically sound, and do the data support the conclusions?

Reviewer #1: Yes

3. Has the statistical analysis been performed appropriately and rigorously? 

Reviewer #1: Yes

4. Have the authors made all data underlying the findings in their manuscript fully available?

Reviewer #1: Yes

5. Is the manuscript presented in an intelligible fashion and written in standard English?

Reviewer #1: Yes

6. Review Comments to the Author

Reviewer #1: Thank you for reviewing, and it is acceptable for publication, because they have answered all the correction questions.

7. PLOS authors have the option to publish the peer review history of their article (what does this mean? ). If published, this will include your full peer review and any attached files.

**Do you want your identity to be public for this peer review?** For information about this choice, including consent withdrawal, please see our Privacy Policy .

Reviewer #1: No

---

## [Editor Report · Acceptance letter]

PONE-D-25-48971R1

PLOS One

Dear Dr. Ying,

I'm pleased to inform you that your manuscript has been deemed suitable for publication in PLOS One. Congratulations! Your manuscript is now being handed over to our production team.

Kind regards,

on behalf of

Mr. Armaan Jamal

Guest Editor

PLOS One